# Weighted gene co-expression network analysis identifies functional modules related to bovine respiratory disease

**Nooshin Ghahramani[1], Ali Hashemi[ID][1]\*, Bahman Panahi[2]**

**1** Department of Animal Science, Division of Animal Breeding and Genetics, Faculty of Agriculture, Urmia University, Urmia, Iran, **2** Department of Genomics, Northwest and West Branch, Agricultural Biotechnology Research Institute of Iran (ABRII), Agricultural Research, Education and Extension Organization (AREEO), Tabriz, Iran

\* a.hashemi50@gmail.com

## Abstract

Bovine respiratory disease *(BRD)* is a multifactorial disease of dairy and beef cattle that involves complex interactions with the host immune system. In the current study, a comprehensive meta-analysis was performed using a *P-value* combination approach. In the next step, the identified meta-genes were subjected to systems biology analysis using the weighted gene co-expression network analysis *(WGCNA)* method. Subsequently, the most functionally important modules and genes were validated using machine learning algorithms. Finally, the critical regulatory network associated with *BRD* was constructed. A total of 1,908 common meta-genes were identified through the combined analysis of differentially expressed genes *(DEGs)* using the *Fisher* and Invorm approaches. Co-expression network analysis confirmed six functional modules, among which the connectivity patterns of the blue, brown, green, and yellow modules were significantly altered in *BRD*-affected cattle compared with healthy controls. Functional enrichment analysis of the significant modules revealed that the *'Salmonella infection,' 'NOD-like receptor signaling pathway,' 'Necroptosis,' 'Toll-like receptor signaling pathway,' 'TNF signaling pathway,' 'IL-17 signaling pathway,' 'Apoptosis,'* and *'Influenza A'* pathways were the most significantly associated with *BRD*. The constructed regulatory network identified *GABPA, TCF4, ELK1, NR2C2,* and *ARNT* as key transcription factors *(TFs)*, each playing a central role in regulating immune and inflammatory pathways implicated in *BRD*. Finally, the constructed model revealed that differential expression of the *CFB* gene is significantly associated with susceptibility to *BRD*. In cattle, *CFB* expression correlates with clinical signs of respiratory disease, supporting its potential as a biomarker. Moreover, the involvement of *CFB* in modulating pro-inflammatory cytokines *(e.g., TNF)* and its integration with other immune-related pathways *(e.g., NF-κB signaling)* further highlight its relevance as a biomarker. Overall, this integrative approach enhances our understanding of the molecular mechanisms underlying *BRD* and provides a

**Data availability statement:** data are available in methods and material section.

**Funding:** The author(s) received no specific funding for this work.

**Competing interests:** The authors have declared that no competing interests exist.

foundation for developing diagnostic, therapeutic, and genetic selection strategies to improve cattle health and disease resistance.

## Introduction

Bovine respiratory disease *(BRD)* is a multifactorial disease involving intricate host-immune interactions influenced by environmental variables and pathogens [1]. The innate immune system represents the first line of defense against *BRD*. Epithelial cells and immune sentinel cells help prevent infection by secreting proinflammatory cytokines. Neutrophils play an essential role in the pathogenesis of *BRD* by promoting inflammation and contributing to lung tissue damage [2]. Environmental factors, including weaning, transportation, overcrowding, and inadequate ventilation, negatively impact the host animal's immune and non-immunological defense mechanisms [3]. Non-immunological defense in *BRD* involves physical and chemical barriers, which act as the first line of defense before the adaptive immune system is activated. These early defense systems play a crucial role in limiting pathogen entry and controlling infection at the onset of disease. *BRD* is commonly associated with bacterial and viral pathogens; The involvement of bacterial species, including *Histophilus somni*, *Mannheimia haemolytica*, and *Mycoplasma bovis*, as well as viruses, such as *bovine respiratory syncytial virus*, *bovine viral diarrhea virus*, and *bovine herpesvirus-1*, has been extensively investigated [4,5]. These bacterial and viral pathogens represent the most prevalent and clinically significant agents implicated in the development and progression of *BRD*. Detecting key genes involved in *BRD* enhances our understanding of the biological pathways that contribute to disease resistance, particularly those related to immune response and inflammation [6,7]. Vaccination programs and treatment of subclinical animals are the primary approaches for preventing and controlling *BRD*. Additionally, identifying genes associated with disease susceptibility can support the development of breeding programs aimed at improving cattle to *BRD* [7].

*RNA sequencing (RNA-Seq)* is a powerful and comprehensive technique used to investigate gene expression profiles, functional mechanisms and molecular processes within distinct biological pathways [8,9]. *RNA-Seq* currently represents the most precise and reliable method for estimating gene expression levels [10]. Several *RNA-Seq* studies have successfully identified key genes and host molecular mechanisms involved in the pathogenesis of *BRD* [1,11,12]. These studies have provided valuable insights into differential gene expression *(DEGs)* patterns, immune response pathways, and regulatory networks that contribute to disease susceptibility and progression [1]. The genes *STAT3, IKBKG, IRAK1, NOD2, TLR2,* and *IKBKB* have been identified as key contributors to the immune response in bovine respiratory disease [13].

Individual *RNA-Seq* studies often face limitations such as small sample sizes, platform variability, and limited statistical power [14]. To address these challenges, meta-analysis provides a robust and standardized approach for integrating data from multiple studies, thereby increasing statistical power and improving the reliability of

gene expression findings [15]. This meta-analysis approach has been used to identify gene expression profiles and evaluate the relative effectiveness of antibiotics in controlling *BRD* in beef cattle [16].

Weighted Gene Co-Expression Network Analysis *(WGCNA)* is a systems biology approach used to identify modules of highly correlated genes [17]. A co-expression network was constructed using *WGCNA* to identify significant modules and potential candidate biomarkers in *BRD* and healthy groups [18].

Functional enrichment analysis identifies specific biological functions that are overrepresented in a group of *DEGs* [19]. The associated functional pathways are obtained from online bioinformatics databases, and the relative abundance of genes relevant to particular pathways is statistically calculated [20].

A gene regulatory network consists of a set of genes that interact to control specific cellular functions. Understanding the relationships between target genes and transcription factor *(TF)*-target interactions provides valuable insights into the organization of these networks and their role in regulating gene expression and cellular processes [21,22].

Machine learning algorithms have become essential tools in genetics due to the their ability to learn patterns from labeled data and make predictions or classifications [23]. Reports indicate that integration co-expression network analysis with gene prioritization using machine learning *(ML)* frameworks is an effective approach for finding new protein functions, cellular and tumor expression profiles, and potential disease-related biomarkers [24,25]. Such approaches have been successfully applied to uncover complex functional regulation and predict expression signatures, providing critical molecular insights into diseases [26]. A subset of *RBD*-related genes, including *PLDA, PHLDA2, VNSC*, and *PAM*, was predicted using a decision tree *(DT)* algorithm [1].

This study aims to construct a gene correlation network based on gene expression data from infected and healthy groups. Furthermore, to validate previous findings obtained through *DEGs* analysis and to potentially identify novel genes and molecular mechanisms associated with *BRD*, *ML* approaches were employed, and candidate biomarkers were proposed for *BRD* prediction.

## Materials and methods

### Data collection

The *RNA-Seq* data associated with *BRD* were obtained from the Gene Expression Omnibus *(GEO)* database (https://www.ncbi.nlm.nih.gov/gds/). Studies were selected based on specific criteria, including recent research on *BRD*, availability of accessible count matrices, clearly defined case and control groups, sufficient sample sizes for robust statistical comparisons, and consistent sequencing platforms. Only studies that utilized whole blood as the biological sample were included, providing consistency in tissue-specific gene expression profiles and minimizing variability introduced by different tissue types. Additionally, the availability of raw count matrices in publicly accessible repositories, such as *GEO*, was a crucial requirement, allowing for independent verification, re-analysis, and integration with other datasets as part of meta-analytical approaches. Five *RNA-Seq* studies related to *BRD* were systematically analyzed. The first dataset **(GSE150706)** profiled the blood transcriptomes of 24 beef steers (n = 72) at three critical stages: **Entry** (on arrival at the feedlot), **Pulled** (when disease is detected), and **Close-out** (recovered, healthy cattle at shipping to slaughter). The goal was to uncover key biological functions, regulatory factors, and gene markers for early diagnosis. Disease identification in the Pulled group was based on a combination of clinical scoring, elevated rectal temperature, and veterinary diagnosis. For analysis, blood transcriptomic data from the Entry and Pulled phases, each comprising 24 samples, were utilized. The second dataset **(GSE152959)** examined the whole blood transcriptomics of Holstein-Friesian calves infected with Bovine Respiratory Syncytial Virus *(BRSV)*, including 6 control calves and 12 infected calves. The third dataset **(GSE162156)** comprised whole blood transcriptomic profiling of heifers without *BRD* (n = 18) and with BRD (n = 25). The fourth dataset **(GSE199108)** included young Holstein-Friesian calves infected with Bovine Herpesvirus 1 *(BoHV-1)* (n = 12) and uninfected calves (n = 6). The fifth dataset **(GSE217317)** examined the relationships between the transcriptome, genome, and *BRD* phenotype of feedlot crossbred cattle using multi-omics analyses, with a total of 143 samples collected from 80 cattle

diagnosed with BRD and 63 pen-matched controls at a single time point. The studies were selected to ensure the reliability of results. Only studies published between 2020 and 2023 were included, representing recent research. To ensure statistical robustness, only studies meeting a predefined minimum sample size were considered. Articles were initially identified using the keywords: *"whole blood RNA-seq," "Bos taurus," "BRD,"* and *"gene expression."* Table 1 summarizes the accession numbers, platforms, sample sizes, layout, and references for the *mRNA-Seq* datasets.

## RNA-Seq data processing

Differentially expressed genes *(DEGs)* between infected and control samples were identified using the *DESeq2* statistical tool (v1.28.1) in the R package with default settings [31]. *DESeq2* utilizes negative binomial generalized linear models to test for *DEGs* and calculates the dispersion for each gene based on its variance and expression level [32]. To employ the *DESeq2* package, the *Ensembl IDs* in each count matrix were converted to *GeneIDs*. The Wald test was applied using *DESeq2* to assess the statistical significance of *DEGs* between conditions. A fold change of ≥|2| and a corrected *P-value* of ≤ 0.05 were used as cutoff points for differential expression [33]. To minimize the impact of batch effects arising from variations in experimental protocols, sequencing platforms, alignment tools, and reference genome annotations, a normalization approach implemented in *DESeq2* was applied. This method effectively reduces unwanted sources of variation commonly encountered in high-throughput sequencing experiments, thereby enhancing the accuracy and comparability of gene expression measurements.

## Meta-analysis of RNA-Seq datasets

Meta-analysis has been widely applied in genetic research, particularly for identifying *DEGs* under two conditions (healthy and infected) in transcriptome analysis [34]. This method has been especially successful in identifying disease-related genes [35]. Meta-analysis encompasses a set of techniques that allow the quantitative combination of data from multiple studies [36]. To minimizes the impact of batch effects arising from variations in experimental protocols, sequencing platforms, alignment tools, and reference genome annotations, we utilized *P-value* combination approaches. Specifically, the *fishcomb* and *invnorm* algorithms were employed to merge *P-values*, as implemented in the meta*RNA-Seq* Bioconductor package (v1.0.5) [37]. Both methods were applied to reduce false positive results, and *DEGs* with a false discovery rate *(FDR)* < 0.05, were considered significant common meta-genes identified by both statistical methods. *Fisher's* method combines the *P-values* from each experiment into a single test statistic defined as follows:

$$x^2 = -2 \sum_{j=1}^{k} \ln (pj)$$

The test statistic χ² follows a χ2 distribution under the null hypothesis, where *pj* indicates the raw *P-value* obtained from genes in the study.

**Table 1. RNA-Seq datasets for transcriptomics analyses of bovine respiratory disease.**

| Accession number | Platform | Samples*(C: T) | Layout | Reference |
|---|---|---|---|---|
| GSE150706 | Illumina HiScanSQ | 24:24 | Paired End | [12] |
| GSE152959 | Illumina NovaSeq 6000 | 6:12 | Paired End | [27] |
| GSE162156 | Illumina NovaSeq 6000 | 18:25 | Paired End | [28] |
| GSE199108 | Illumina NovaSeq 6000 | 5:12 | Paired End | [29] |
| GSE217317 | Illumina HiSeq 4000 | 41: 59 | Paired End | [30] |

## Weighted gene co-expression network analysis

The *WGCNA* Bioconductor R package (v3.5.1) was used to detect correlation patterns among genes and identify significant modules across *RNA-Seq* datasets. Network construction; module identification; module and gene selection; calculation of network topological features; visualization was all performed using the *WGCNA* approach. The expression values of meta-genes were normalized using the variance-stabilizing transformation *(vst)* function [38]. To increase the reliability of the constructed co expression network, an outlier detection step to remove unusual samples that could bias the results. After removing outliers, the number of applied samples exceeded 15, further enhancing the robustness of the study. Additionally, a signed method was applied for network construction. Pearson correlation was used for both outlier detection and meta-gene identification due to its computational efficiency [39]. Meta-genes were identified based on pairwise Pearson correlation coefficients. The relatively large sample size in this study also contributed to the robustness and reliability of the findings. Subsequently, the similarity matrix was converted into an adjacency matrix. The corresponding dissimilarity matrix *(1 − TOM)* and topological overlap matrix *(TOM)* were derived from adjacency matrix [17]. Finally, a dynamic hybrid tree-cutting technique was employed to identify modules, with average linkage hierarchical clustering performed using the topological overlap-based dissimilarity matrix as input [40].

## Functional enrichment analysis

To investigate the functional significance of the identified modules, enrichment analysis was performed using Gene Ontology *(GO)* and Kyoto Encyclopedia of Genes and Genomes *(KEGG)* via the *STRING* and *DAVID* databases [41]. This analysis identifies biological-processes *(BPs),* molecular-functions *(MFs),* and cellular-components *(CCs)* that are significantly overrepresented. Detecting these enriched biological and molecular pathways provides valuable insights into the underlying biological mechanisms of *BRD*.

## Inferring gene regulatory networks

To infer the regulatory network of significant modules within the constructed co-expression network, *Cytoscape* software was used to map *(TF)*-target interactions. By constructing gene regulatory networks, key regulatory genes associated with specific diseases, such as *BRD,* can be identified. The *iRegulon* plugin in *Cytoscape* was employed to identify these regulators and their corresponding *TF*-target interactions. Subsequently, regulatory relationships between the top target genes and the identified *TFs* were constructed and visualized, providing a comprehensive view of how these molecular components interact to influence disease-related pathways and processes.

## Supervised machine-learning models

Supervised machine learning involves training a model on a dataset that includes input features (genes) and known outputs (healthy or disease status). The machine learning process can be divided into several key stages: data pre-processing, data splitting, model selection, model training, and model evaluation [42]. We employed machine learning algorithms to identify and validate the most significant meta-genes associated with *BRD*, ensuring precise gene prioritization with optimal accuracy. Feature selection algorithms based on weighting methods were applied to prioritize key genes related to *BRD*. Hub genes from significant modules were used to select important features using 10 distinct weighting algorithms, including information gain, information gain ratio, χ2, deviation, rule, support vector machine, Gini index, uncertainty, relief, and principal component analysis *(PCA)* [43]. Next, the selected features were used to optimize predictive modeling with the Decision Trees *(DTs)* algorithm. To evaluate the model's accuracy, a 10-fold cross-validation approach was applied. In this method, the data were partitioned into 10 equal subsets; in each iteration, 9 subsets were used for training and 1 subset for testing, ensuring thorough assessment of the model's generalizability and predictive performance. Ultimately, the model with the highest accuracy was proposed as the best predictor for identifying *BRD*-associated genes.

## Results

### Meta-analysis

The overall workflow of our systems biology approach is summarized in Fig 1.

As shown in Fig 2, the number of *DEGs* varies across datasets.

The meta-analysis results (see Fig 3) indicate that 1908 common meta-genes were identified.

### Weighted gene co-expression network construction

To obtain a reliable network and identify significant modules with high correlation, outlier samples were removed from the initial datasets. Preliminary analysis revealed that samples with a standard connectivity score *(Z.k)* lower than −2.5 were excluded, and the remaining samples were used for weighted co-expression network construction. After removing outliers, the high-quality data were subjected to *WGCNA* to explore underlying gene expression patterns. The soft-thresholding power *(β)* was selected by systematically evaluating the scale-free topology fit index *(R²)* across a range of candidate

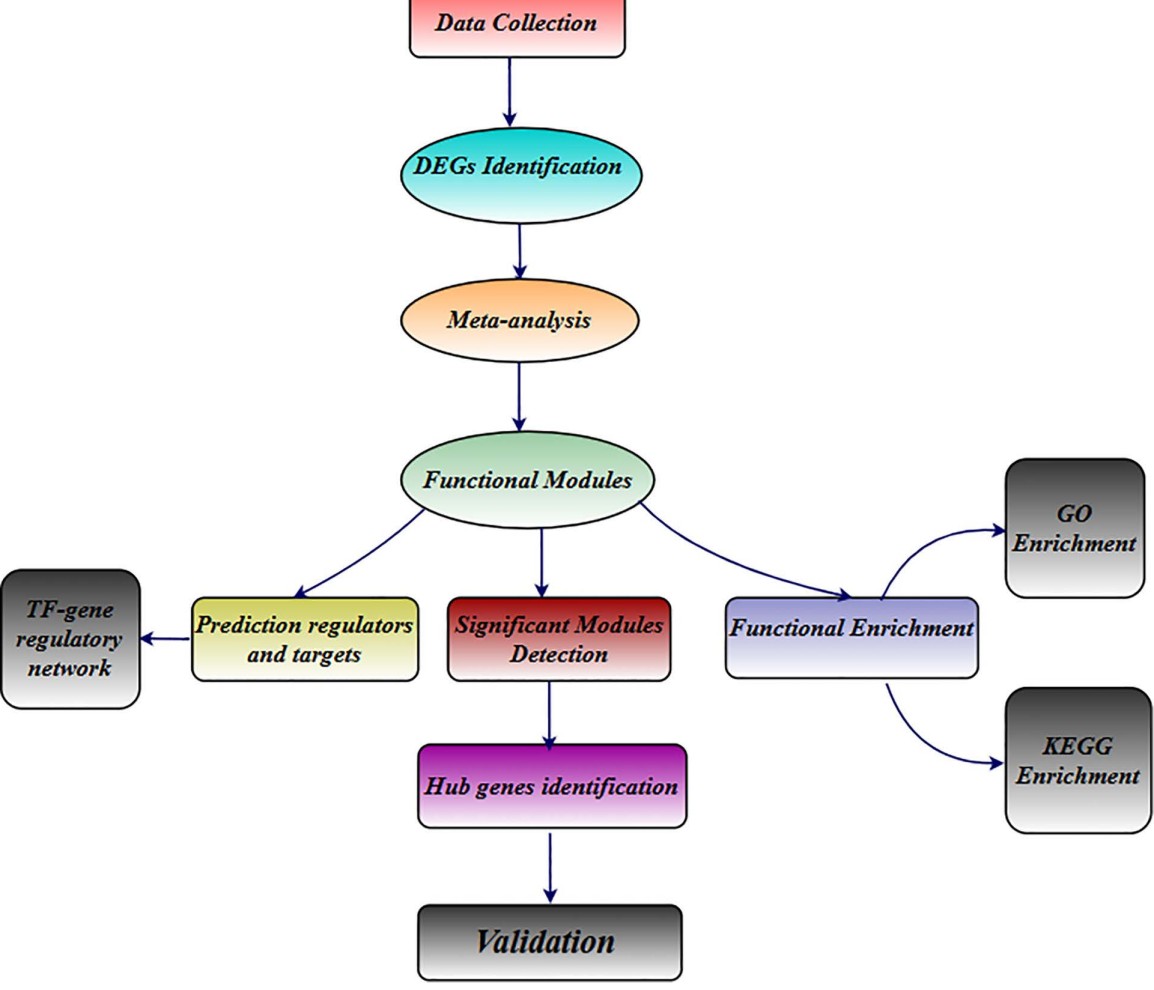

**Fig 1. A step-by-step workflow of the systems biology approach used in this project.** The five RNA-Seq datasets comprise a total of 226 independent samples, including 132 BRD-infected samples and 94 control samples from dairy and beef cattle experiments.

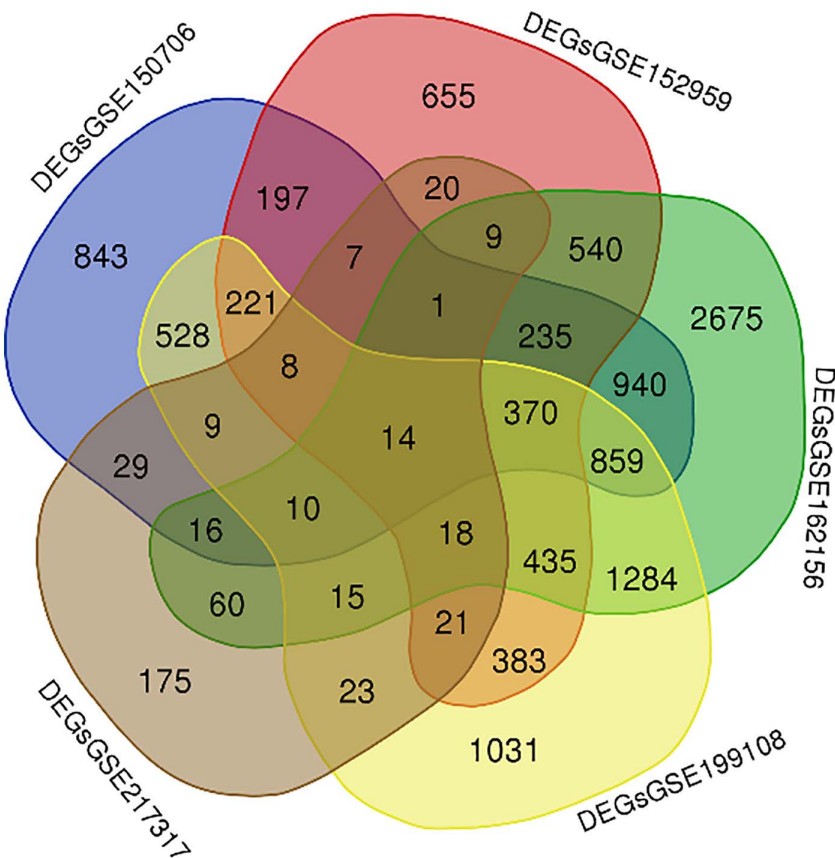

**Fig 2. The number of DEGs identified in each dataset: 4287 in GSE150706, 3134 in GSE152959, 7481 in GSE162156, 5229 in GSE199108, and 435 in GSE217317.**

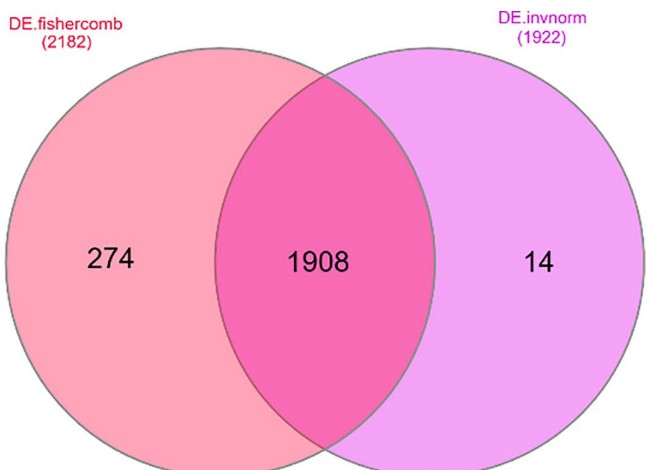

**Fig 3. Overall, 2182 and 1922 meta-genes were identified in response to BRD using the Fisher and Invorm methods, respectively.** Results of the meta-analysis showing 1908 common meta-genes identified by both methods for subsequent gene correlation analysis.

powers. The optimal *β* was defined as the smallest power at which the network exhibited approximate scale-free properties *(R²>0.8)*. This selection ensures that the constructed network reflects the expected biological architecture of gene co-expression, thereby enhancing the accuracy and interpretability of subsequent module detection and functional analysis. Identification of six co-expressed gene modules with an average size of 318 genes is shown in Fig 4A. The size distribution of the co-expressed gene modules is summarized in Fig 4B. The heat map of the *TOM* illustrating gene interconnectedness within the modules is shown in Fig 4C**.** Module eigengenes representing expression patterns within each module are shown in Fig 4D**.**

### Functional enrichment analysis of meta-genes

The results of functional enrichment analysis of significant modules revealed 525, 58, and 57 *GO* terms for *BPs, MFs,* and *CCs,* respectively. The most critical terms enriched in the *BPs* included *"Response to stress," Regulation of response to stimulus," Regulation of metabolic process," "Positive regulation of biological process," and "Regulation of immune response,".* In the *CCs* category, significantly enriched terms included *"Cytoplasm," "Intracellular organelle," "Nucleus," "Nucleoplasm,"* and *"Cytoplasmic vesicle".* The most significantly enriched *BPs* and *CCs* identified from the analyzed gene modules are shown in Fig 5.

   In the *MFs* category, significant enrichment of binding-related terms was observed, including *'binding,' 'protein binding,' 'enzyme binding,' 'ATP binding,'* and *'small molecule binding.'* This enrichment indicates substantial involvement of molecular interactions that are critical for various biochemical and regulatory processes. Additional details related to *MFs* can be found in Table 2, which offers further insights and relevant data supporting the analysis.

   The most over-represented *KEGG* pathways identified within the significant modules included the *'NOD-like receptor signaling pathway,' 'TNF signaling pathway,' 'IL-17 signaling pathway,' 'NF-kappa B signaling pathway,'* and *'T cell receptor signaling pathway.'* These pathways are known to play critical roles in immune response and inflammation. Comprehensive details regarding these significant modules, including associated genes and enrichment statistics, are provided in Table 3 to support further interpretation and analysis.

### *TFs*- hub genes regulatory network

There are 537, 154, 288 and 506 target genes in the blue, brown, green and yellow modules, respectively. The transcriptional regulatory network of the main target genes and *TFs* was established using *Cytoscape.* Key target genes were identified based on the intra-modular connectivity criterion. The *TFs GABPA, TCF4, ELK1, NR2C2* and *ARNT* were identified as important regulators, controlling 24, 14, 20, and 22 hub genes in the blue, brown, green, and yellow modules, respectively. These *TFs* specifically regulate the hub genes within the significant modules associated with *BRD*. A large number of *TFs* can regulate hub genes, and their expression may reflect the complexity of mechanisms that lead to *BRD*. *TCF4* was found to directly interact with the *STK40, PRKAB1, MAP3K11,* and *OGFR* genes in the brown, blue, yellow and green modules, respectively. Key *TFs* interacting with target genes within the constructed regulatory networks are shown in Fig 6.

### Validation hub genes in co-expressed modules

Initially, normalized gene expression values were assigned to the data matrix, which served as the foundational input for subsequent analyses. The *DT* method was employed to confirm the identified hub genes using four distinct criteria: information gain, accuracy, Gini index, and information gain ratio. According to the data, the *DT* using the gain ratio criteria achieved the highest accuracy (70%). Based on the expression levels of meta-genes, the *DT* model confirmed the functional significance of the top-ranked genes in categorizing respiratory diseases in cattle. The *DT* highlighting the *CFB* gene as a potential biomarker for *BRD* is shown in Fig 7. Samples were classified as *BRD* when the expression levels of

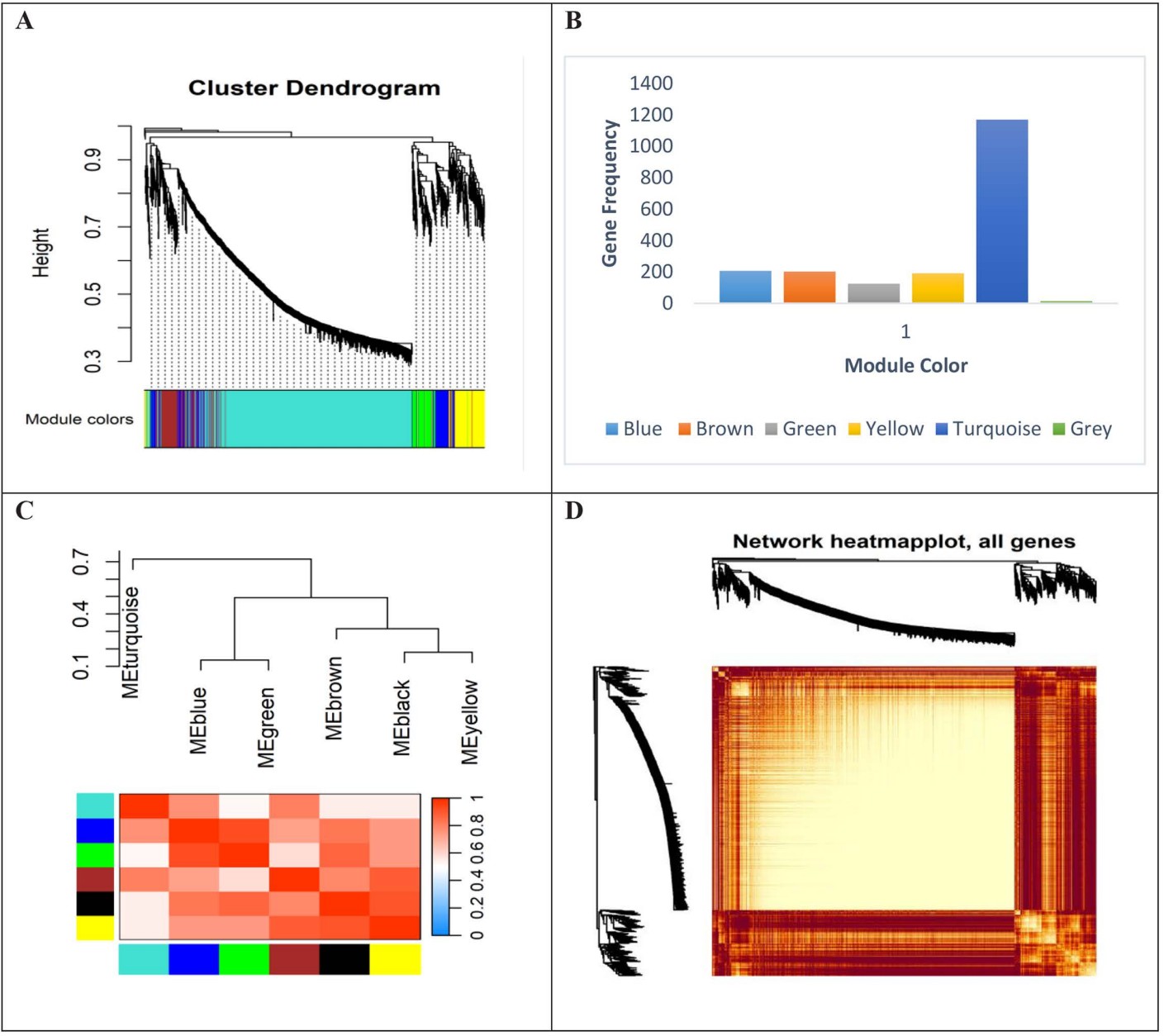

**Fig 4. A. Identification of six co-expressed gene modules with an average size of 318.** B. Size distribution of the co-expressed gene modules, highlighting the turquoise module as the largest (1169 genes) and the gray module as the smallest (14 genes). Genes unrelated to any module were assigned to the gray module. The brown (n = 202 genes), yellow (n = 191 genes), blue (n = 207 genes), and green (n = 125 genes) modules were identified as critical functional modules associated with BRD through WGCNA analysis. C. Heat map of the TOM showing the degree of interconnectedness among genes within the modules identified by the dynamic tree cutting algorithm. Yellow and progressively red colors indicate low and high TOM values, respectively. D. Module eigengenes representing the gene expression patterns within each module, derived as the first principal component of each module's expression data matrix.

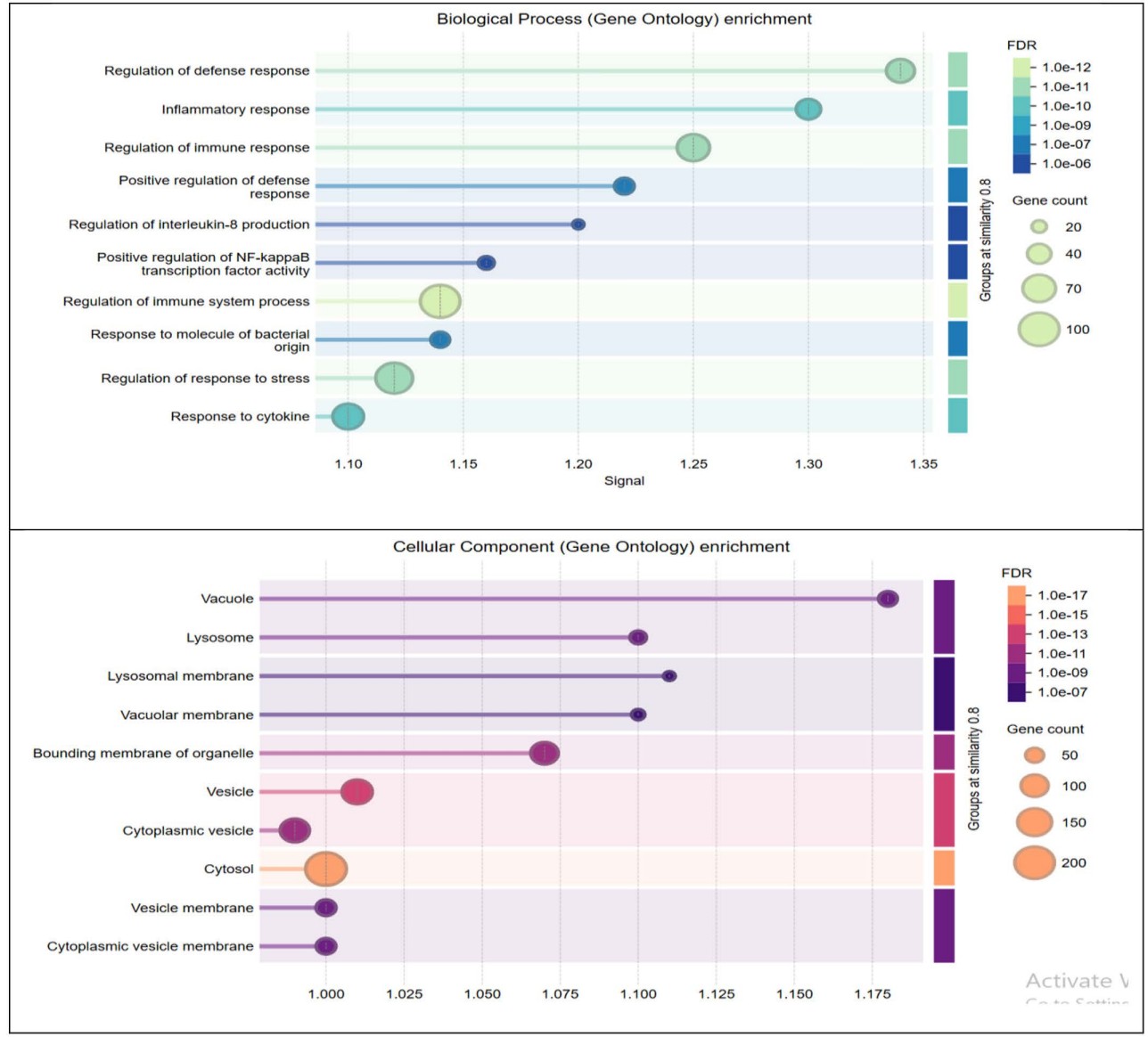

**Fig 5. Most significantly enriched BPs and CCs terms identified from the analyzed gene modules.** These terms highlight the main biological roles and cellular localizations involved in BRD.

*CFB* and *STAT2* were greater than 131.500 and 9161, respectively. Alternatively, samples were also identified as *BRD* when *CFB* expression was > 131.5, *STAT2* ≤ 9161, *NLRC5* ≤ 11922, *IL15RA* ≤ 800, and *ALPK1* > 635. These findings indicate that *CFB* expression consistently plays a central role in *BRD* classification, depending on the expression context of other genes such as *STAT2, NLRC5, IL15RA*, and *ALPK1*.

## Discussion

Complex host immune interactions in dairy and beef cattle during *BRD* are influenced by pathogenic agents and environmental factors [44]. Additionally, the inclusion of animals from naturally infected and experimentally challenged studies

**Table 2. Significant MF terms of meta-genes in bovine respiratory disease.**

| Term ID | Term description | P-value |
|---|---|---|
| GO:0019899 | Enzyme binding | 1.43E-14 |
| GO:0005515 | Protein binding | 7.23E-10 |
| GO:0042802 | Identical protein binding | 1.27E-07 |
| GO:0019901 | Protein kinase binding | 7.44E-06 |
| GO:0005488 | Binding | 8.88E-06 |
| GO:0003824 | Catalytic activity | 1.65E-05 |
| GO:0019900 | Kinase binding | 3.12E-05 |
| GO:0044389 | Ubiquitin-like protein ligase binding | 6.26E-05 |
| GO:0016772 | Transferase activity, transferring phosphorus-containing groups | 0.0008 |
| GO:0043168 | Anion binding | 0.00082 |
| GO:0019904 | Protein domain specific binding | 0.00099 |
| GO:0030554 | Adenyl nucleotide binding | 0.00099 |
| GO:0043167 | Ion binding | 0.0012 |
| GO:0140096 | Catalytic activity, acting on a protein | 0.0012 |
| GO:0016301 | Kinase activity | 0.0015 |
| GO:0005524 | ATP binding | 0.0017 |
| GO:0016740 | Transferase activity | 0.0017 |
| GO:0016773 | Phosphotransferase activity, alcohol group as acceptor | 0.0023 |
| GO:0036094 | Small molecule binding | 0.0031 |
| GO:0030674 | Protein-macromolecule adaptor activity | 0.0055 |

*Only the significantly enriched (p < 0.05) GO terms are presented.*

may introduce variability in gene expression profiles. The complex regulation of these genes may correspond to the enrichment of specific biological pathways. However, due to limited sample sizes in some individual studies, the statistical power to detect reliable gene-level changes was insufficient. To overcome this limitation and enhance the reliability of the results, a meta-analysis approach was employed. By combining transcriptome datasets across multiple studies, this method increased the overall analytical power and provided new insights into the relationships and expression patterns of key regulatory genes associated with *BRD*. Furthermore, using the *Fisher* and *Invorm* methods, our analysis enabled more meaningful and robust identification of meta-genes. These outcomes were subsequently used in system biology analysis to characterize patterns of gene correlation across samples. Modules of highly connected genes were then subjected to pathway enrichment analysis. Finally, gene regulatory networks between *TFs* and target genes were inferred using *Cytoscape* to elucidate effective genes involved in the development of *BRD*. Supervised *ML* algorithms were selected for this study, as the primary objective was to predict and validate genes based on known biological patterns and labeled data. Overall, the meta-analysis approach detected 1908 common meta-genes using the *Fisher* and *Invorm* methods. System biology analysis revealed that the topological characteristics of the *BRD* gene co-expression network differed, resulting in the formation of four co-expressed modules: blue, green, yellow and brown. These modules were selected for further investigation.

The analysis of co-expression modules revealed key genes potentially involved in *BRD*. In the blue module, the most significant genes included *PRCC, CHMP7, ABCF3, BAG6,* and *ADCK2*; the brown module included *TBC1D14, ELL, STK40, NOD2,* and *CYRIA*; the yellow module featured *MAP3K11, LRP10, LRPAP1, SPI1,* and *GPAT4*; and the green module contained *OGFR, GET3, ADRM1, MUL1,* and *WASHC1*. The *PRCC* gene has been implicated in signaling cascades that may contribute to tumorigenesis in lung cancer [45,46]. The *CHMP7* gene has been associated

**Table 3. The significant KEGG metabolic pathways associated with meta-genes.**

| Pathway name | P-value | Total genes in pathway | Strength |
|---|---|---|---|
| Salmonella infection | 5.15E-14 | 38 | 0.77 |
| NOD-like receptor signaling pathway | 9.56E-10 | 28 | 0.74 |
| Osteoclast differentiation | 1.45E-08 | 22 | 0.79 |
| PD-L1 expression and PD-1 checkpoint pathway in | 5.15E-08 | 19 | 0.83 |
| Cancer | 1.12E-07 | 22 | 0.73 |
| Yersinia infection | 1.14E-07 | 23 | 0.7 |
| Necroptosis | 1.38E-07 | 17 | 0.85 |
| Leishmaniasis | 3.70E-07 | 16 | 0.85 |
| Adipocytokine signaling pathway | 8.02E-07 | 24 | 0.62 |
| Kaposi sarcoma-associated herpesvirus infection | 1.28E-06 | 18 | 0.74 |
| NF-kappa B signaling pathway | 1.32E-06 | 17 | 0.76 |
| Toll-like receptor signaling pathway | 1.32E-06 | 23 | 0.62 |
| Tuberculosis | 2.43E-06 | 18 | 0.71 |
| TNF signaling pathway | 6.25E-06 | 19 | 0.65 |
| Apoptosis | 7.02E-06 | 21 | 0.6 |
| Influenza A | 9.09E-06 | 20 | 0.61 |
| Hepatitis B | 9.59E-06 | 15 | 0.74 |
| IL-17 signaling pathway | 9.59E-06 | 19 | 0.63 |
| Measles | 1.01E-05 | 23 | 0.55 |
| Epstein-Barr virus infection | 1.32E-05 | 16 | 0.69 |
| Chagas disease | 1.99E-05 | 16 | 0.67 |
| Toxoplasmosis | 2.26E-05 | 13 | 0.77 |
| Pertussis | 2.65E-05 | 22 | 0.54 |
| Human immunodeficiency virus 1 infection | 2.65E-05 | 22 | 0.54 |
| Viral carcinogenesis | 3.86E-05 | 12 | 0.78 |
| Mitophagy – animal | 5.23E-05 | 11 | 0.81 |
| Legionellosis | 8.37E-05 | 18 | 0.56 |
| Hepatitis C | 8.53E-05 | 25 | 0.46 |
| MAPK signaling pathway | 8.53E-05 | 11 | 0.78 |
| Cytosolic DNA-sensing pathway | 8.53E-05 | 14 | 0.66 |

with immunodeficiency, translation regulation [47], membrane deformation, cell division, metabolism, and development; notably, its low expression may impair these biological processes [48]. *BAG6*, a novel regulatory protein, was found in porcine respiratory syndrome virus [49]. *TBC1D14,* detected in our study, was introduced as a novel metastasis biomarker by downregulating macrophage-erythroblast interactions [50]. The *ELL* gene connects the regulation of transcription elongation to cell growth [51]. *STK40* was revealed as a key gene whose deletion alters the expression of genes important for lung development [52]. Disruption of *NOD2* gene has been associated with impaired resistance to inflammatory diseases [53], while its interaction with *BRSV* suggests a potential role in the innate immune response to viral infection [54]. *MAP3K11* was described as a target gene in lung cancer [55], contributing to the proliferation of activated epithelial cells [56]. *SPI1* and *PSTPIP1* were discovered as novel early prognostic biomarkers relevant to innate immune response in lung adenocarcinoma [57,58]. Further, *OGFR* [59] and *GET3* play critical roles in regulating immunological functions, influencing both innate and adaptive immune responses. The immune system plays a key role in defending the body against pathogens, and maintaining cellular homeostasis. Regulating immune functions supports controlled inflammation, enhances antioxidant defense, facilitates cell signaling and repair, and promotes the removal

**Fig 6. Key transcription factors ELK1, GABPA, NR2C2, and ARNT interacting directly with target genes PMM2, ACAA1, MAFG, and TMED1 within the constructed regulatory networks.**

of damaged cells [60]. According to our findings, *ADRM1* plays a role in the migration, survival, and proliferation of cancer cells [61]. *MUL1* was identified as a novel regulator of antiviral response, limiting inflammation, and downregulation mitochondrial respiration [62].

Functional enrichment analysis of the significant modules indicated that these modules were enriched in *BPs* such as *"Regulation of defense response," "Inflammatory response," "Regulation of immune response," "positive regulation of biological processes"* [12], and *"Regulation of signal transduction"* [63] in *BRD*. Additionally, consistent with our results, several important *MFs*, including *"signal transduction", "Protein binding," "enzymes binding,"* [64,65] *"ion binding,"* and *"catalytic activity"* [40] have also been detected in bacterial respiratory infections in calves during *BRD* [66]. Analysis of *KEGG* pathways in the significant modules revealed that the co-expressed genes were highly enriched in pathways such as *"Salmonella infection", "NOD-like receptor signaling pathway", "Toll-like receptor signaling pathway", "Apoptosis", "IL-17 signaling pathway", "Tumor Necrosis Factor"* and *"NF-kappa B signaling pathway"*. *"Salmonella infection"* has been shown to affect cattle, horses, and pigs causing reproductive losses and clinical disease such as *BRD*. Several previous studies indicated that the *"NOD-like receptor signaling pathway," "Toll-like receptor signaling pathway,"* and *"Apoptosis"* are enriched in diverse respiratory bacterial and viral pathogens [67]. Our findings also revealed that the key genes in

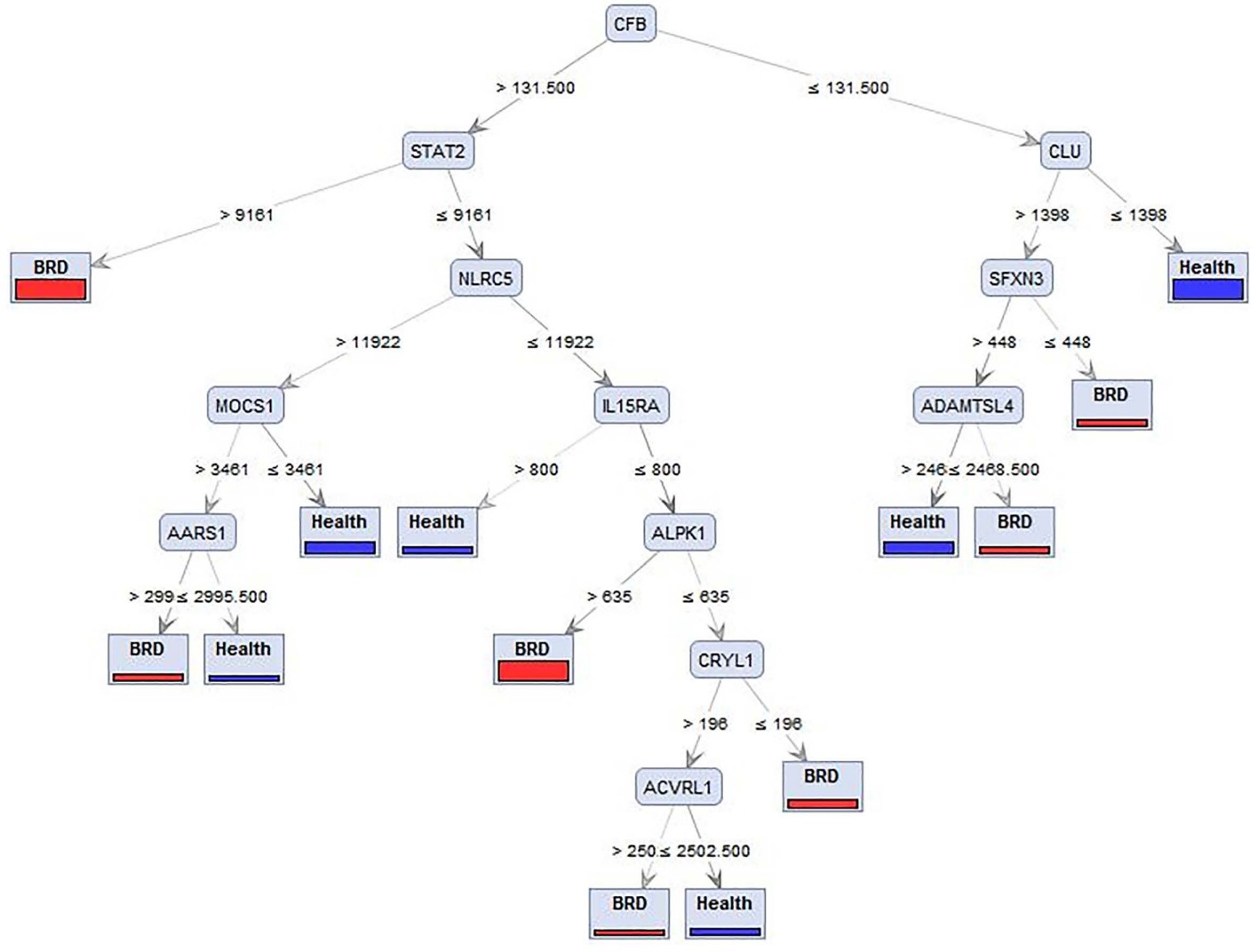

**Fig 7. DT highlighting the CFB gene as a potential biomarker for BRD, located at the root of the tree.**

significant modules regulated the *"IL-17 signaling pathway," "Tumor Necrosis Factor,"* and *"NF-kappa B signaling path-way,"* all of which are related to immune processes. Notably, similar pathway associations have been previously reported in studies of *BRD* [9,13]. The target genes and regulatory networks of *TFs* associated with *BRD* recurrence were identified, including 25 *TFs,* of which five key *TFs* were selected to construct a regulation network. Target genes and *TFs* with high connective degrees in the regulatory networks are reported to be associated with *BRD*. Our results indicated that the *TFs GABPA, TCF4, ELK1, NR2C2* and *ARNT* are strongly associated with *BRD.* In line with our findings, the functions of *GABPA* gene have been reported in the pathogenesis of leukemia and cancer [68–70]. *TCF4* is a key *TF* involved in the regulation of Wnt/β-catenin signaling during lung development [71]. *Elk1* has been shown to increases the inflammatory response and exacerbates lung damage in acute respiratory syndrome by controlling the transcription of *Fcgr2b* [72]. The expression of the *NR2C2* gene, a potential target of *miR-378b,* was significantly upregulated in lung cancer tissues [73]. In addition, module preservation analysis identified *NR2C2* as a hub gene in *BRD* [63].

BRD is influenced by both genetic and non-genetic factors. Genetic factors include inherited traits such as immune response and stress resilience, which may predispose animals to either susceptibility or resistance to the disease. Non-genetic factors involve environmental and management-related conditions, such as transportation stress, weaning, weather changes, and overcrowding. Given the importance of BRD in cattle, certain genes are expected to show considerable expression changes during the disease process. Previous studies employing ML algorithms in genetic analyses have successfully identified key genes associated with various diseases. Therefore, we applied supervised ML techniques, which enabled the accurate identification of key genes in cattle affected by BRD. Our analysis revealed that attribute weighting algorithms consistently highlighted the following genes: CFB, CLU, STAT2, NLRC5, MOCS1, IL15RA, CRYL1, SFXN3, ADAMTSL4, AARS1, ALPK1, and ACVRL1. These genes are primarily involved in inflammation, immune system function, and cellular proliferation and differentiation. Some of the studies included in this analysis were challenge experiments in which BRD was experimentally induced. Therefore, findings related to innate susceptibility to BRD were interpreted with caution. Among the identified genes, CFB, which regulates cellular senescence, was observed to be upregulated, consistent with previous reports of its role as an immune-related gene during BRD [28]. As indicated by the DT model, CLU functions as a tumor suppressor in the early stages of carcinogenesis, and has been suggested as a relevant gene in lung cancer [74]. STAT2 plays a critical role in immune responses and acts as a TF regulating the expression of numerous genes involved in host defense mechanisms during BRD [12]. MOCS1 participates in reactions that produce an essential cofactor and has been identified as a candidate causative mutation for bovine disease [75]. Additionally, our findings confirmed differential expression of IL15RA, CRYL1, and ALPK1 during BRD. IL15RA was identified as a potential marker for genetic resistance to infection in cattle [76]. The CRYL1 gene product catalyzes the dehydrogenation of L-gulonate into dehydro-L-gulonate, and its increased expression has been related to heat stress in dairy cows [77], suggesting a possible connection between metabolic stress and BRD development. ALPK1 triggers activation of the inflammatory NF-κB signaling pathway and plays a vital role in the pathogenesis of lung cancer [78]. Through bioinformatics analysis of RNA-sequencing data, we successfully identified key genes associated with BRD. Furthermore, we developed a diagnostic model using a supervised ML algorithm to predict potential molecular markers for BRD. This study provides valuable insights into the molecular mechanisms of BRD and contributes to the identification of potential diagnostic markers.

## Conclusions

The integrative systems biology analysis conducted in this study revealed several key metabolic processes and signaling pathways involved in BRD. Notably, critical target genes were identified and used to construct regulatory networks with TFs. By employing a supervised ML algorithm in combination with gene clustering approaches, this study provides comprehensive insights into significant genes, immune system components, and molecular functions associated with BRD. These findings not only enhance our understanding of the underlying biological mechanisms but also establish a valuable foundation for future research aimed at clinical diagnosis, biomarker identification, and the discovery of novel therapeutic targets.

## Author contributions

**Conceptualization:** Ali Hashemi, Bahman Panahi.

**Data curation:** Nooshin Ghahramani, Ali Hashemi.

**Formal analysis:** Nooshin Ghahramani.

**Visualization:** Nooshin Ghahramani.

**Writing – original draft:** Nooshin Ghahramani, Ali Hashemi.

**Writing – review & editing:** Ali Hashemi, Bahman Panahi.

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
