## [Decision Letter · Decision Letter 0]

9 Jun 2025

PONE-D-25-08222Weighted gene co-expression network analysis identifies functional modules related to bovine respiratory diseasePLOS ONE

Dear Dr. Hashemi,

Thank you for submitting your manuscript to PLOS ONE. After careful consideration, we feel that it has merit but does not fully meet PLOS ONE’s publication criteria as it currently stands. Therefore, we invite you to submit a revised version of the manuscript that addresses the points raised during the review process.

Both reviewers have raised important concerns regarding the manuscript that require extensive revisions for we to consider it further. Please address all the reviewers' comments when/if revising the manuscript. Note the attached file with comments from one of the reviewers. Also, the English language of the manuscript is not at a stage where it would be considered suitable for publication. Please work to revise the language throughout the manuscript for grammar and syntax correctness. 

We look forward to receiving your revised manuscript.

Kind regards,

Angel Abuelo, DVM, MRes, MSc, PhD, DABVP (Dairy), DECBHM

Academic Editor

PLOS ONE

Journal Requirements:

https://journals.plos.org/plosone/article?id=10.1371%2Fjournal.pone.0307248

In your revision ensure you cite all your sources (including your own works), and quote or rephrase any duplicated text outside the methods section. Further consideration is dependent on these concerns being addressed.

[Funding support of Urmia university are gratefully being knowledge.]

[The author(s) received no specific funding for this work.]

6. Please include a separate caption for each figure in your manuscript.

Reviewers' comments:

Reviewer's Responses to Questions

**Comments to the Author**

1. Is the manuscript technically sound, and do the data support the conclusions?

Reviewer #1: Yes

Reviewer #2: No

2. Has the statistical analysis been performed appropriately and rigorously? 

Reviewer #1: Yes

Reviewer #2: I Don't Know

3. Have the authors made all data underlying the findings in their manuscript fully available?

Reviewer #1: Yes

Reviewer #2: Yes

4. Is the manuscript presented in an intelligible fashion and written in standard English?

Reviewer #1: Yes

Reviewer #2: No

5. Review Comments to the Author

Reviewer #1: Thank you for the opportunity to review the manuscript "Weighted gene co-expression network analysis identifies functional modules related to bovine respiratory disease" by Hashemi and colleagues. The investigators of this work sought to perform a meta-analysis of previously published RNA-Seq data specifically focused on bovine respiratory disease. I commend the investigators on their use of previously published literature/data and more novel bioinformatic analysis methods in an effort to corroborate and identify molecular features associated with BRD outcomes. I believe the paper to be technically sound and the results impactful, however a few areas require additional clarity and elaboration prior to publication. Below are my suggestions and questions which may improve the work prior to publication.

Line 12 (abstract): The authors highlight their use of a "P-value combination approach" regarding a brief summarization of their methods for the abstract. This is somewhat ambiguous, as their methodology in a combinational approach of WCGNA, machine learning workflows, and statistical methods (ex: DESeq2). The lack of specificity in this sentence may distract future readers finding their abstract in literature searches; they may consider removing this part of the sentence and instead emphasize the aspect of the meta-analysis approach (including the number of studies included).

Line 39-42 (and throughout): minor typographical errors and run-on sentences were identified. For example, it is believed that "through" should be "thorough". Additionally, it is unclear how the "identification of key genes" would prevent and control BRD; it is believed that the authors are eluding to how these identifications would lead to the development/employment of early disease detection systems.

Line 78-79: it is unclear why the investigators specifically elected to use these five studies. There are additional published studies that have used whole blood as the tissue sample of evaluation (assumed as the connecting theme of these works), but the authors should expand upon the rationale behind these studies. The utility and approach behind performing a meta-analysis-like study in bioinformatics remains a highly contested subject, so it is assumed there are no strict criteria as to how to perform a study of this type. However, the authors should provide context and rationale when possible as this work may be evaluated in the future as a way of performing this type of study (outside of BRD or animal health research).

Table 1, citation 20: this citation should be revised; DOI: 10.1016/j.ygeno.2020.07.014

Lines 95-118, methods: the authors utilized previously published RNA-Seq data for their statistical and combinational (p-value) analyses. However, I would express concern regarding the possibility of batch effects/technical bias across these five studies, as the laboratory techniques and bioinformatic processing, including alignment tools and reference assembly, are different across these projects (ex. Sun et al., 2020 utilized TopHat2 and the UMD3.1 genome, Johnston et al., 2021 utilized STAR and the UMD3.1 genome, and the work from the Jiminez, O’Donoghuem and Li publications utilized STAR and the UCD1.2 genome). The way this section is currently written, it is assumed the investigators acquired and combined the raw gene counts from each of these publications (not explicitly stated). The authors are recommend to comment and elaborate on their approach, as without the re-processing of combined raw data nor the use of a batch effect adjustment of raw gene counts (see DOI: 10.1093/nargab/lqaa078), this project risks presenting confounded results.

Lines 100-102: which testing procedure was used with DESeq2 (i.e., Wald testing, likelihood ratio testing, etc.)?

Line 125: the use of Pearson correlation coefficients with WGCNA (and generally within statistics) are often considered to be limited in interpretation due to the sensitivity to outlier data and risk of estimator bias, especially when applied to data with limited sample size. While the authors of WGCNA have often recommended to have a minimum of 15 samples (well surpassed by this study), they also recommend a more robust coefficient (such as the biweight midcorrelation) and signed networking for adjacency matrix construction. While this technique has been used and published previously, the investigators may comment on the rationale of this aspect of their methods.

Lines 130-142: which platform/software/toolkit was used to perform functional enrichment analyses? The authors explain the databases used, but not the specific toolkit (ex. DAVID, WebGestalt, gProfiler, etc.).

Lines 154-163: similar to the previous comment, it is unclear what program was utilized here. Additionally, the techniques described are supervised algorithms which require training and cross-validation. Therefore, a confusion matrix should be produced from these algorithms which would have demonstrated the classification statistics (i.e., F1 scores, AUCs, balanced accuracies, recall, precision, etc.). It is assumed that the authors utilized the balanced accuracies as their determination for model selection, but this should be explicitly stated and the full statistics of all models should be provided as a table or supplemental file.

Line 231: the values used in the decision tree are not described. Are these adjusted gene count values or do they represent something else?

Figures: it may be the formatting within the PDF file, but please ensure that the figures are of high quality/resolution. Specifically, Figure 3 is blurred and the "Fishercomb" header is clipped from the image.

Reviewer #2: • The authors should consult a native English speaker to improve the readability of this article.

• The authors have a well-organized methods section but should be sure to include sufficient detail for the methods being described. The authors should describe how they searched for datasets to include in their meta-analysis and what inclusion/exclusion criteria were applied. Additionally, the authors should justify their methodological choices (e.g., line 178: removing samples with a standard connectivity score less than -2.5).

• The overall structure/organization of the discussion had logical flow. Within each paragraph of the discussion, I encourage the authors to condense and summarize the information rather than discussing each point individually. Coalescing the relevant information into a single overarching statement with better contextualizing the relevance of each outcome on BRD would improve the readability of the discussion.

• In general the authors should familiarize themselves with current research on biomarkers in animal health and BRD to be able to more appropriately interpret their findings.

6. PLOS authors have the option to publish the peer review history of their article (what does this mean? ). If published, this will include your full peer review and any attached files.

**Do you want your identity to be public for this peer review?** For information about this choice, including consent withdrawal, please see our Privacy Policy .

Reviewer #1: **Yes: ** Matthew A. Scott

Reviewer #2: No

---

## [Author Response · Author response to Decision Letter 1]

26 Jul 2025

Reviewer #1: •

1- Line 12 (abstract): The authors highlight their use of a "P-value combination approach" regarding a brief summarization of their methods for the abstract. This is somewhat ambiguous, as their methodology in a combinational approach of WCGNA, machine learning workflows, and statistical methods (ex: DESeq2). The lack of specificity in this sentence may distract future readers finding their abstract in literature searches; they may consider removing this part of the sentence and instead emphasize the aspect of the meta-analysis approach (including the number of studies included).

Dear reviewer, Thank you for your insightful consideration of our manuscript. in current study, comprehensive meta-analysis was performed using P-value combination approach, in the next step, the identified meta genes were subjected the systems biology analysis using weighted gene co-expression network method. Then, the most functionally importance modules and genes were further validated using machine-learning approach. Finally, the critical regulatory network associated with BRD were constructed.

2- Line 39-42 (and throughout): minor typographical errors and run-on sentences were identified. For example, it is believed that "through" should be "thorough". Additionally, it is unclear how the "identification of key genes" would prevent and control BRD; it is believed that the authors are eluding to how these identifications would lead to the development/employment of early disease detection systems.

Dear reviewer, Thank you for your insightful consideration of our manuscript. has been extensively investigated (3, 4). Detecting key genes involved in BRD, enables a better understanding of the biological pathways that contribute to disease resistance, particularly those related to immune response and inflammation (5, 6).

3- Line 78-79: it is unclear why the investigators specifically elected to use these five studies. There are additional published studies that have used whole blood as the tissue sample of evaluation (assumed as the connecting theme of these works), but the authors should expand upon the rationale behind these studies. The utility and approach behind performing a meta-analysis-like study in bioinformatics remains a highly contested subject, so it is assumed there are no strict criteria as to how to perform a study of this type. However, the authors should provide context and rationale when possible as this work may be evaluated in the future as a way of performing this type of study (outside of BRD or animal health research).

Dear reviewer, Thank you for your insightful consideration of our manuscript. these studies were selected based on several key criteria to ensure the reliability and reproducibility of the results. Only studies that utilized whole blood as the biological sample were included, as this provides consistency in tissue-specific gene expression profiles and minimizes variability introduced by different tissue types. This allows for independent verification, re-analysis, and integration with other datasets as part of meta-analytical approaches. Table 1, citation 20: this citation should be revised; DOI: 10.1016/j.ygeno.2020.07.014. It was revised, please see the citations in the Table 1

4- Lines 95-118, methods: the authors utilized previously published RNA-Seq data for their statistical and combinational (p-value) analyses. However, I would express concern regarding the possibility of batch effects/technical bias across these five studies, as the laboratory techniques and bioinformatic processing, including alignment tools and reference assembly, are different across these projects (ex. Sun et al., 2020 utilized TopHat2 and the UMD3.1 genome, Johnston et al., 2021 utilized STAR and the UMD3.1 genome, and the work from the Jiminez, O’Donoghuem and Li publications utilized STAR and the UCD1.2 genome). The way this section is currently written, it is assumed the investigators acquired and combined the raw gene counts from each of these publications (not explicitly stated). The authors are recommending to comment and elaborate on their approach, as without the re-processing of combined raw data nor the use of a batch effect adjustment of raw gene counts (see DOI: 10.1093/nargab/lqaa078), this project risks presenting confounded results.

Dear reviewer, Thank you for your insightful consideration of our manuscript. to minimize the impact of batch effects resulting from variations in experimental protocols, sequencing platforms, alignment tools, and reference genome annotations, we first used a normalization approach implemented in the DESeq2 software. This method effectively reduces unwanted sources of variation commonly encountered in high-throughput sequencing experiments, thereby enhancing the accuracy and comparability of gene expression measurements.

Lines 100-102: which testing procedure was used with DESeq2 (i.e., Wald testing, likelihood ratio testing, etc.)?

Dear reviewer, Thank you for your insightful consideration of our manuscript. The Wald test was calculated using DESeq2 to assess the statistical significance of DEGs between conditions.

5- Line 125: the use of Pearson correlation coefficients with WGCNA (and generally within statistics) are often considered to be limited in interpretation due to the sensitivity to outlier data and risk of estimator bias, especially when applied to data with limited sample size. While the authors of WGCNA have often recommended to have a minimum of 15 samples (well surpassed by this study), they also recommend a more robust coefficient (such as the biweight midcorrelation) and signed networking for adjacency matrix construction. While this technique has been used and published previously, the investigators may comment on the rationale of this aspect of their methods.

Dear reviewer, thank you for your critical reviewing of our manuscript. As you mentioned to increase the reliability of constructed co expression network, we performed the outlier detection step to remove the outliers from the study, moreover, after the removing the outliers the applied samples we higher than 15 samples, which increase the reliability of our study, furthermore, the applied approach were signed method for network construction.

6- Lines 130-142: which platform/software/toolkit was used to perform functional enrichment analyses? The authors explain the databases used, but not the specific toolkit (ex. DAVID, WebGestalt, gProfiler, etc.).

Dear reviewer, Thank you for your insightful consideration of our manuscript. To investigate the functional significance of the identified modules, enrichment analysis was carried out based on Gene Ontology (GO) and Kyoto Encyclopedia of Genes and Genomes (KEGG) using the STRING and DAVID databases (39). This procedure detects biological-processes, molecular-functions, and cellular-components that are crucially over-represented. Detecting over-represented biological and molecular pathways donates valued comprehension of biological mechanisms in BRD.

7- Lines 154-163: similar to the previous comment, it is unclear what program was utilized here. Additionally, the techniques described are supervised algorithms which require training and cross-validation. Therefore, a confusion matrix should be produced from these algorithms which would have demonstrated the classification statistics (i.e., F1 scores, AUCs, balanced accuracies, recall, precision, etc.). It is assumed that the authors utilized the balanced accuracies as their determination for model selection, but this should be explicitly stated and the full statistics of all models should be provided as a table or supplemental file.

Dear reviewer, Thank you for your insightful consideration of our manuscript. Supervised machine learning involve training a model on a dataset that includes both input features (genes) and known outputs (healthy or disease status). The process involved in machine learning can be broken down into several key stages: data pre-processing, data splitting, model selection, model training, and model evaluation (41). We employed machine learning algorithms to identify and validate the most significant meta-genes associated with BRD, ensuring precise gene prioritization with optimal accuracy.

9- Line 231: the values used in the decision tree are not described. Are these adjusted genes count values or do they represent something else?

Dear reviewer, Thank you for your insightful consideration of our manuscript. the provided values are the normalized expression values of genes.

8- Figures: it may be the formatting within the PDF file, but please ensure that the figures are of high quality/resolution. Specifically, Figure 3 is blurred and the "Fishercomb" header is clipped from the image.

Dear reviewer, Thank you for your insightful consideration of our manuscript. we revised and increased the quality of figure. Please see the revision format.

Reviewer #2: • The authors should consult a native English speaker to improve the readability of this article.

Dear reviewer, Thank you for your insightful consideration of our manuscript. we focused on improving the readability and logical flow of the article.

• The authors have a well-organized methods section but should be sure to include sufficient detail for the methods being described. The authors should describe how they searched for datasets to include in their meta-analysis and what inclusion/exclusion criteria were applied. Additionally, the authors should justify their methodological choices (e.g., line 178: removing samples with a standard connectivity score less than -2.5).

Dear reviewer, thanks you for your insightful comments, we utilized p-value combination approaches for meta-analysis. as described in methods and material section. To minimize the impact of batch effects arising from variations in experimental protocols, sequencing platforms, alignment tools, and reference genome annotations, normalization and batch effect reduction approach were performed.

For WGCNA, Following the removal of outlier samples, the remaining high-quality data were subjected to weighted gene co-expression network analysis to explore the underlying gene expression patterns. The soft-thresholding power (β) was selected by systematically evaluating the scale-free topology fit index (R²) across a range of candidate powers. The optimal β was defined as the smallest power at which the network began to exhibit approximate scale-free properties (R² > 0.8). This selection process ensures that the constructed network reflects the expected biological architecture of gene co-expression, thereby improving the accuracy and interpretability of subsequent module detection and functional analysis. Moreover, the heat map illustrated the TOM values to visualize the degree of interconnectedness among genes within the network modules identified using the dynamic tree cutting algorithm (Fig 4C). Yellow and progressively red colors demonstrate the low and higher TOM values, respectively. The gene expression patterns within each module were condensed into a single representative vector, named the module eigengene, which corresponds to the first principal component derived from the module’s expression data matrix (Fig 4D).

• The overall structure/organization of the discussion had logical flow. Within each paragraph of the discussion, I encourage the authors to condense and summarize the information rather than discussing each point individually. Coalescing the relevant information into a single overarching statement with better contextualizing the relevance of each outcome on BRD would improve the readability of the discussion.

Dear reviewer, Thank you for your insightful consideration of our manuscript. It was revised. Please see the revision.

• In general the authors should familiarize themselves with current research on biomarkers in animal health and BRD to be able to more appropriately interpret their findings.

Dear reviewer, Thank you for your insightful consideration of our manuscript. the CFB gene, which regulates cellular senescence, was observed to be upregulated in this study, consistent with previous findings suggesting its role as an immune-related gene during BRD

---

## [Decision Letter · Decision Letter 1]

24 Sep 2025

PONE-D-25-08222R1Weighted gene co-expression network analysis identifies functional modules related to bovine respiratory diseasePLOS ONE

Dear Dr. Hashemi,

Thank you for submitting your manuscript to PLOS ONE. After careful consideration, we feel that it has merit but does not fully meet PLOS ONE’s publication criteria as it currently stands. Therefore, we invite you to submit a revised version of the manuscript that addresses the points raised during the review process. Please see below some minor change suggestions from the reviewers.

We look forward to receiving your revised manuscript.

Kind regards,

Angel Abuelo, DVM, MRes, MSc, PhD, DABVP (Dairy), DECBHM

Academic Editor

PLOS ONE

Journal Requirements:

Reviewers' comments:

Reviewer's Responses to Questions

**Comments to the Author**

1. If the authors have adequately addressed your comments raised in a previous round of review and you feel that this manuscript is now acceptable for publication, you may indicate that here to bypass the “Comments to the Author” section, enter your conflict of interest statement in the “Confidential to Editor” section, and submit your "Accept" recommendation.

Reviewer #1: All comments have been addressed

Reviewer #2: (No Response)

2. Is the manuscript technically sound, and do the data support the conclusions?

Reviewer #1: Yes

Reviewer #2: Partly

3. Has the statistical analysis been performed appropriately and rigorously? 

Reviewer #1: Yes

Reviewer #2: N/A

4. Have the authors made all data underlying the findings in their manuscript fully available?

Reviewer #1: Yes

Reviewer #2: Yes

5. Is the manuscript presented in an intelligible fashion and written in standard English?

Reviewer #1: Yes

Reviewer #2: Yes

6. Review Comments to the Author

Reviewer #1: Thank you for the opportunity to review the revised manuscript. The author's have addressed all comments and made substantial revisions to the manuscript.

Reviewer #2: Thank you for addressing the changes so thoroughly. The authors have done a great job improving the discussion by focusing on the results relevant to BRD and contextualizing within their model.

Line 29: For consistency please remove the alpha from TNF

Line 95-97: I appreciate that the authors have elaborated on their selection process, however i think the this description still needs to be expanded further. As written another researcher does not have sufficient detail to reproduce the article selection. Could the authors include the range of dates that they considered recent research, what was the sample size cut off required for inclusion or exclusion, as well as the keywords used to initially identify articles.

Line 118: The description of table 1 still includes species, please remove as that has been removed from the table.

Line 326: I caution the authors when discussing oxidative stress not to over interpret their gene expression data. The authors should avoid statements such as “effectively protecting cells against oxidative protein damage” as they did not assess if cells were protected against oxidative protein damage.

7. PLOS authors have the option to publish the peer review history of their article (what does this mean? ). If published, this will include your full peer review and any attached files.

**Do you want your identity to be public for this peer review?** For information about this choice, including consent withdrawal, please see our Privacy Policy .

Reviewer #1: No

Reviewer #2: No

---

## [Author Response · Author response to Decision Letter 2]

30 Sep 2025

Editor,

review your reference list to ensure that it is complete and correct.

Dear editor, we carefully reviewed all the references from the beginning and re-imported them using EndNote. Some references were duplicates and were removed, while others contained errors, which we also eliminated. We also thoroughly reviewed the entire manuscript for grammar and syntax and applied the necessary corrections.

Line 29: For consistency, please remove the alpha from TNF

Dear reviewer, we removed the α from TNF

Line 95-97: I appreciate that the authors have elaborated on their selection process, however i think this description still needs to be expanded further. As written another researcher does not have sufficient detail to reproduce the article selection. Could the authors include the range of dates that they considered recent research, what was the sample size cut off required for inclusion or exclusion, as well as the keywords used to initially identify articles.

Dear reviewer, we restricted our search to studies published between 2020 and 2023, which we considered representative of recent research. To ensure statistical robustness, only studies meeting a predefined minimum sample size were included. Articles were initially identified using the following keywords: “whole blood RNA-seq,” “Bos taurus,” “BRD,” and “gene expression.”"

We also reviewed the first part of the Materials and Methods section and applied the necessary corrections.

Line 118: The description of table 1 still includes species, please remove as that has been removed from the table.

Dear reviewer, we removed species from the text.

Line 326: I caution the authors when discussing oxidative stress not to over interpret their gene expression data. The authors should avoid statements such as “effectively protecting cells against oxidative protein damage” as they did not assess if cells were protected against oxidative protein damage.

Dear reviewer, oxidative stress removed from the text.

---

## [Editor Report · Decision Letter 2]

2 Oct 2025

Weighted gene co-expression network analysis identifies functional modules related to bovine respiratory disease

PONE-D-25-08222R2

Dear Dr. Hashemi,

We’re pleased to inform you that your manuscript has been judged scientifically suitable for publication and will be formally accepted for publication once it meets all outstanding technical requirements.

Kind regards,

Angel Abuelo, DVM, MRes, MSc, PhD, DABVP (Dairy), DECBHM

Academic Editor

PLOS ONE
---

## [Editor Report · Acceptance letter]

PONE-D-25-08222R2

PLOS ONE

Dear Dr. Hashemi,

I'm pleased to inform you that your manuscript has been deemed suitable for publication in PLOS ONE. Congratulations! Your manuscript is now being handed over to our production team.

Kind regards,

on behalf of

Dr. Angel Abuelo

Academic Editor

PLOS ONE